# Bacteriophages M13 and T4 Increase the Expression of Anchorage-Dependent Survival Pathway Genes and Down Regulate Androgen Receptor Expression in LNCaP Prostate Cell Line

**DOI:** 10.3390/v13091754

**Published:** 2021-09-02

**Authors:** Swapnil Ganesh Sanmukh, Nilton José dos Santos, Caroline Nascimento Barquilha, Maira Smaniotto Cucielo, Márcio de Carvalho, Patricia Pintor dos Reis, Flávia Karina Delella, Hernandes F. Carvalho, Sérgio Luis Felisbino

**Affiliations:** 1Laboratory of Extracellular Matrix Biology, Department of Structural and Functional Biology, Institute of Biosciences of Botucatu, Sao Paulo State University (UNESP), Botucatu 18618-689, SP, Brazil; ssanmukh@ibecbarcelona.eu (S.G.S.); nilton.unesp@gmail.com (N.J.d.S.); caroline.barquilha@gmail.com (C.N.B.); maira.cucielo@gmail.com (M.S.C.); flavia.delella@unesp.br (F.K.D.); 2Department of Structural and Functional Biology, Institute of Biology, University of Campinas (UNICAMP), Campinas 13083-970, SP, Brazil; hern@unicamp.br; 3Department of Surgery and Orthopedics, Faculty of Medicine, Sao Paulo State University (UNESP), Botucatu 18618-687, SP, Brazil; marcio.carvalho@unesp.br (M.d.C.); patricia.reis@unesp.br (P.P.d.R.)

**Keywords:** prostate cancer, gene expression, integrin, bacteriophage, nanoparticle

## Abstract

Wild-type or engineered bacteriophages have been reported as therapeutic agents in the treatment of several types of diseases, including cancer. They might be used either as naked phages or as carriers of antitumor molecules. Here, we evaluate the role of bacteriophages M13 and T4 in modulating the expression of genes related to cell adhesion, growth, and survival in the androgen-responsive LNCaP prostatic adenocarcinoma-derived epithelial cell line. LNCaP cells were exposed to either bacteriophage M13 or T4 at a concentration of 1 × 10^5^ pfu/mL, 1 × 10^6^ pfu/mL, and 1 × 10^7^ pfu/mL for 24, 48, and 72 h. After exposure, cells were processed for general morphology, cell viability assay, and gene expression analyses. Neither M13 nor T4 exposure altered cellular morphology, but both decreased the MTT reduction capacity of LNCaP cells at different times of treatment. In addition, genes *AKT*, *ITGA5*, *ITGB1*, *ITGB3*, *ITGB5*, *MAPK3*, and *PI3K* were significantly up-regulated, whilst the genes *AR*, *HSPB1*, *ITGAV*, and *PGC1A* were down-regulated. Our results show that bacteriophage M13 and T4 interact with LNCaP cells and effectively promote gene expression changes related to anchorage-dependent survival and androgen signaling. In conclusion, phage therapy may increase the response of PCa treatment with *PI3K/AKT* pathway inhibitors.

## 1. Introduction

Prostate Cancer (PCa) is the second leading cause of death in men globally [1,2]. PCa is curable in most cases if detected before causing distant metastasis to bone and other body organs [3,4]. Nevertheless, earlier detection and diagnosis must be accompanied by new treatment options for advanced stages, which are resistant to anti-androgenic and/or chemotherapies [5,6,7].

Bacteriophages have been used as an alternative to antibiotics. Phage-based therapies, as well as phage lytic-enzyme therapies, have been reported. The effects of phages on cancer cells in vitro as well as in vivo are promising [8,9,10,11,12]. Similarly, hybrid phages or modified phages have been developed by various groups worldwide to specifically detect, target, and attack different cancer cells including PCa [10,13,14,15].

Bacteriophages can utilize mammalian viruses’ route for their entry into the cells as they have been found inside endosomes, lysosomes, Golgi, cytoplasm, and the nucleus of mammalian cells [16,17] and to kill intracellular bacteria [18]. Hence, the involvement of bacteriophages in-activation of Toll-like receptors and alteration of immune cell response is evident [19,20]. Moreover, phage genome is reported to activate the expression of host genes after reaching the mammalian cell nucleus [21,22,23], which in still less understood and needs further investigation.

The mechanism by which phages can enter mammalian cells includes phage uptake through phagocytosis [16], but it seems to differ for different types of phages [17]. The T4 phage is reported to transcytose into different organs through their epithelial cell barriers [24], whereas M13 phages can enter both epithelial and endothelial cells through different types of endocytosis or macropinocytosis [25]. Additionally, mammalian cell/tissue types as well as the size of phage particles affect the rate of uptake [17,26,27]. However, in vivo uptake is not yet characterized and needs directed attention [16].

As the advancement in the field of phage-based biomedical techniques, as well as genetically engineered phages in human phage therapy [28], are on the rise, it is very important to know how these nano-bio particles interact with mammalian cells in their natural forms. In this sense, it seems mandatory to evaluate their potential interactions with cancer cells to understand their possible effects. However, few reports explore the effect of non-modified phages on cancer cell lines in vitro for the understanding of gene expression changes associated with cell proliferation, cell growth, and/or cell death [15,29,30,31,32].

Here, we report important findings associated with the interaction of bacteriophages M13 and T4 with the LNCaP prostate cancer cell line. We think that this is the first report showing the direct effect of bacterial viruses on crucial factors affecting prostate cancer cell progression. We have assessed their effects on cell viability and genes important for cancer cell growth and proliferation (*AR*, *AKT*, *PI3K*, *MAPK*, *HSP90*, *HSPB1*, *PGC1A*), adhesion, migration, and invasion (integrins *ITGA5*, *ITGAV*, *ITGB1*, *ITGB3*, *ITGB5*, and *ACTB*). The results suggest that treatment affects cell metabolism, turning cells less dependent on the *AR* signaling and more dependent on the *AKT, PI3K/MAPK* pathways, which are easily druggable, suggesting the possibility for the use of phages in combination therapies.

## 2. Materials and Methods

### 2.1. LNCaP Cell Culture

LNCaP cells (ATCC CRL-1740) were purchased from American Type Cell Culture (Manassas, VA, USA). The LNCaP cells were cultured initially in a 25 cm^2^ culture flask (Qiagen, Crawley, West Sussex, UK) with RPMI 1640 medium (Gibco/Thermo Fisher Scientific, Waltham, MA, USA) supplemented with 10% fetal bovine serum (FBS) (Gibco/Thermo Fisher Scientific, Waltham, MA, USA), 50 μg/mL penicillin, 50 μg/mL streptomycin and 0.5 μg/mL amphotericin B (Gibco/Thermo Fisher Scientific, Waltham, MA, USA), and were incubated in a CO_2_ incubator at 37 °C until 90% confluency. The cells were subjected to 0.05% trypsin (Gibco/Thermo Fisher Scientific, Waltham, MA, USA), and transferred to a 75 cm^2^ culture flask (Qiagen, Crawley, West Sussex, UK). All the experiments and analyses were carried out as per the standard procedures and following the guidelines provided by respective authorities. The medium was changed every 2 days, and cells were daily monitored using an inverted microscope (Zeiss Axiovert-Oberkochen, Baden, Württemberg, Germany). When cells reached 90% confluency, they were subjected to 0.05% trypsin and transferred to a new culture flask or to 24/6-well culture plates (Corning, Corning, NY, USA) to initiate the different treatments and experiments.

### 2.2. LNCaP Cell Exposure to Bacteriophages M13 and T4

Bacteriophages M13KE (New England Biolabs Inc., Ipswich, MA, USA) and Coliphage T4 (T4r+) (Carolina Inc., Burlington, UK) were acquired purified at 10^11^ pfu/mL in peptone broth and were prepared as per published protocols [9]. The phages were not expanded in bacterial culture to avoid LPS or endotoxin contamination. The phages were centrifuged at 10,000× *g* for 30 min followed by filtration through 0.22 μm cellulose acetate membrane filter (Millipore™). Phage preparations were diluted 100-fold with phosphate buffered saline (1 × PBS) to reach a concentration of ~10^9^ pfu/mL. The phages were further diluted in cell culture medium to reach 1 × 10^5^, 1 × 10^6^, and 1 × 10^7^ pfu/mL for treating the cells. This series of dilutions significantly reduced the peptone broth components in the cell culture medium and any possible trace of LPS or endotoxin.

### 2.3. MTT Reduction Assay

LNCaP cells (6 × 10^4^) were seeded in 24-well plates. Once cells became 70% confluent, they were treated with bacteriophages M13 and T4 (10^7^ pfu/mL). For this experiment, we chose the highest concentration of phages to check the effect on cell viability. After 4, 24, and 48 h of exposure, cell viability was determined by the MTT (Thiazolyl Blue Tetrazolium Bromide-Sigma-Aldrich, St. Louis, MO, USA) reduction method according to the manufacturer’s instructions [33,34]. The reaction was transferred to a 96-well plate and read in a spectrophotometer (ASYS HITECH GmbH, Eugendorf, Salzburg-Umgebung, Austria) at 550 nm to determine the percentage of cell viability relative to control cells.

### 2.4. Hematoxylin and Eosin Staining

The LNCaP cells were grown in 12-well plates containing coverslips on the bottom. After reaching 30% of confluency, LNCaP cells were exposed to the vehicle (PBS) or the highest concentration of bacteriophage M13 or T4 treatment at 10^7^ pfu/mL for 24, 48, and 72 h, washed in PBS and fixed with 10% formaldehyde in PBS for 30 min. Cells were washed in PBS and stained by hematoxylin-eosin. The coverslips were dried in ethanol, mounted in a glass slide with Permount, and observed in a Leica DMLB microscope (Leica Inc., Wetzlar, Germany).

### 2.5. RNA Extraction and cDNA Synthesis for qPCR Studies

The LNCaP cells were exposed to bacteriophage M13 or T4 treatment for 24, 48, and 72 h at 1 × 10^5^, 1 × 10^6,^ and 1 × 10^7^ pfu/mL. For total RNA extraction, the culture medium was aspirated, and cells were washed with PBS. Total RNA was extracted using the All-Prep DNA/RNA/Protein extraction kit (Qiagen, Crawley, West Sussex, UK), according to the manufacturer’s instructions. The total RNA extracted was quantified using NanoVue (GE Healthcare, Chicago, IL, USA). Two micrograms of total RNAs were reverse transcribed using the high-capacity RNA-to-cDNA kit (Life Technologies, Carlsbad, CA, USA) in a 20 μL reaction according to the manufacturer’s instructions.

For Real-Time PCR, the Power SYBR Green/ROX qPCR Master Mix reagent (2 ×) (Applied Biosystems, Waltham, MA, USA) was used. The total reaction volume per sample was 10 μL (5.0 μL Power SYBR Green, 0.8 μL of each forward and reverse oligonucleotides (Table 1) [800 nM], 3.2 μL nuclease-free water, 1 μL cDNA), performed in triplicate using 384-well plates. The reaction was performed in the QuantStudio 12K Flex thermal cycler (Applied Biosystems, Waltham, MA, USA), and the results were evaluated by QuantStudio 12K Flex Real-Time PCR System v1.1 program. The reaction consisted of the following cycling: step 1 at 50 °C for 2 min and 95 °C for 2 min; step 2 at 95 °C for 1 s, and step 3 at 60 °C for 30 s; steps 2 and 3 were repeated 40 times; dissociation curve with incubation at 95 °C for 15 s and 60 °C for 1 min with subsequent increase in temperature from 60 °C to 95 °C at the rate of 0.15 °C per second.

For the calculation of gene expression, the ΔΔCt method was used [35], which is based on the exponential PCR reaction, according to the formula QR = 2^–ΔΔCt^, where QR represents the level of gene expression; Ct represents the amplification cycle in which each sample exhibits exponential amplification; ΔCt refers to the difference between the Ct of the amplified sample for the target gene and the Ct of the same amplified sample for the reference gene, and ΔΔCt represents the difference between the ΔCt of the sample of interest at a given time and the ΔCt of the reference sample.

The fold change was calculated from 2^–∆∆Ct^ and then Log_2_FoldChange was calculated. The results were shown as log_2_FoldChange (log_2_FC).

The reactions were performed in triplicate for 14 target genes *ACTB*, *AKT*, *AR*, *HSPB1*, *HSP90*, *ITGA5*, *ITGAV*, *ITGB1*, *ITGB3*, *ITGB5*, *MAPK1*, *MAPK3*, *PGC1A*, and *PI3K*, with *GAPDH* as an endogenous control in a Real-Time PCR System AB7900 (Applied Biosystems, Waltham, MA, USA), according to the manufacturer’s instructions. The values for all samples are normalized by the ratio obtained between the target gene and the mean Cts obtained for the reference gene *GAPDH*. Forward and reverse primers are listed in Table 1.

### 2.6. Statistical Analysis

The results presented in the heatmap were expressed in log_2_FoldChange (log_2_FC). The Shapiro–Wilk normality test was used for normal data distribution. For results that did not pass the normality test, the Kruskal–Wallis non-parametric test was used with Dunn’s Multiple Comparison Test. Differences were considered statistically significant when *p* < 0.05. Statistical analyses were performed using GraphPad Prism (version 5.00, Graph Pad, Inc., San Diego, CA, USA). The results were shown as a heatmap prepared with the Morpheus tool (https://software.broadinstitute.org/morpheus/ (accessed on 25 August 2021)).

## 3. Results

### 3.1. Cell Morphology

LNCaP cells exposed to bacteriophages M13 and T4 at 1 × 10^7^ pfu/mL showed no significant morphological alterations after 24 h, as compared to untreated cells (Figure 1). Similar results were observed after 48 and 72 h with both M13 and T4 phages.

### 3.2. MTT Reduction-Cell Viability

The exposure to bacteriophages temporarily reduced the viability of prostate cancer cells. After 4 and 24 h of treatment, the bacteriophage M13 decreased the viability of LNCaP cells by 23% and 30%, respectively. The bacteriophage T4 reduced the cell’s viability by 29% after 4 h of exposure. There was no significant difference in LNCaP viability after 48 h of treatment with both M13 and T4 bacteriophage (Figure 2).

### 3.3. Gene Expression Profiles after Exposure to Bacteriophages M13 and T4

Gene expression analysis of LNCaP cells after exposure to M13 phage has shown integrin genes up-regulated. After 24 h, 48 h, and 72 h of exposure to the M13 phage, the genes *ITGA5*, *ITGB1*, *ITGB3*, and *ITGB5* showed an increase in gene expression when compared to untreated cells. Furthermore, M13 treatments also increased the gene expression of *AKT*, *MAPK3*, and *PI3K*. However, important genes such as *AR*, *ITGAV*, *HSPB1*, and *PCG1A* were down-regulated after treatment with M13 phage in LNCaP cells. No changes in the cytoskeleton protein *ACTB* gene expression were observed (Figure 3A).

Gene expression of LNCaP cells after T4 phage exposure showed similar expression changes. After exposure to T4, the genes *AKT*, *ITGA5*, *ITGB1*, *ITGB3*, *ITGB5*, *MAPK3*, and *PI3K* were up-regulated. Similarly, the genes *AR*, *HSPB1*, *ITGAV*, and *PGC1A* were down-regulated. No changes in the cytoskeleton protein *ACTB* gene expression were observed (Figure 3B).

## 4. Discussion

Considering the importance of phage peptides in regulating important genes within cancer cells and tumors, as reported in the last few decades [36,37,38,39,40], it is important to investigate how wild-type bacteriophages interact with human cells, particularly due to their role in influencing the immune system [8,9,41].

Our results demonstrate the effects of phage M13 and T4 interaction with the LNCaP cell line by interfering with a set of genes important for cancer cell progression. Curiously, despite remarkable changes in gene expression, we observed no significant change in cellular morphology, after hematoxylin-and-eosin staining. As phages are modulating integrin genes expression, their role in the regulation of cell shape (cytoskeleton), as well as cell migration cannot be ignored [42].

The results of LNCaP cell viability assays indicate that M13 and T4 phages affect the viability of LNCaP cells following binding, as previously suggested [43,44]. It is recently demonstrated that the internalization of phages by normal mammalian cells is responsible for the potential sink of phages during both in vitro as well as in vivo phage applications [17,18,21,22,23]. Further experimental approaches should be undertaken to verify if the process of phage internalization is also occurring in LNCaP cells and how this could interfere, therapeutically, for targeting prostate cancer.

Downregulation of *AR* following phage treatments is an interesting finding. AR is a master regulator of prostate epithelial cell proliferation and function. In particular, AR regulates differentiation of the prostate epithelial cell, controlling the expression of genes such as *KLK3* (PSA) and *PSMA*, two important markers of prostate differentiation. Despite the AR downregulation in LNCaP cells after bacteriophage exposure, tumor cell viability was only transiently compromised after 4 and 24 h of treatment for T4 and M13 phages, respectively.

Integrins, *PI3K*, and *AKT* were upregulated after phage treatment. We hypothesize that these pathways are involved in sustaining cellular survival and growth, as a compensatory mechanism for AR signaling suppression [45,46]. Inhibition of PI3K/AKT/mTOR is a well-established target for cancer therapy, including prostate cancer [47,48,49,50]. Thus, considering that M13 and T4 changed the gene expression profile of LNCaP cells and have effects on anchorage-dependent survival, our results suggest that phage peptides can be used in PCa treatment combined with other adjuvant therapies, such as PI3K/AKT inhibitors. Interpreting the mechanism by which natural phages alter such cancer cell progression genes can be of great importance as they are available along with normal microflora in our body and can be responsible for various conditions previously not understood in cancer patients [31,51].

We also observed downregulation of *HSPB1* in LNCaP cells following the treatment of phage M13 and T4. *HSPB1* gene encodes one of the Small Heat Shock Proteins (sHSP) related to HSP27 (also known as mammalian sHSP family or HSPB family). It is reported that high levels of *HSPB1* in prostate cancer are associated with poor clinical outcomes, as *HSPB1* expression results in tumor invasion and metastasis [52,53]. Similarly, it was found that downregulation of HSP27 (*HSPB1*) in MCF-7 human breast cancer cells upregulates *PTEN*, which is responsible for apoptosis [53]. Additionally, clinical trials showed that the binding of RP101 (brivudine) to *HSPB1* increases survival in both experimental animals and pancreatic cancer patients [54].

Similarly, in the initial 24 h of treatment of LNCaP cells with M13 and T4 phages, the PGC1α expression was significantly decreased. *PGC1α* has been reported to control prostate cancer growth and metastasis [41]. *PGC1α* activates an estrogen-related receptor alpha (*ERRα)*-dependent transcriptional program to elicit a catabolic state and suppression of metastasis. It has been observed that the PGC1α-ERRα pathway exhibited prognostic potential in prostate cancer, thereby contributing to disease stratification and treatment [55]. It has been reported that synthetic androgen (R1881) increases *PGC1α* mRNA expression. The observed variation in *PGC1α* indicates that phages are valuable candidates and must be considered for their potential in the treatment of advanced PCa [56]. We also observed a similar effect on PC3 cancer cells as their interaction with these bacteriophages negatively affected cell migration and growth, using our pipette tip gap closure migration assay (s-ARU method) [57]. Nonetheless, the transient decrease in cell proliferation/viability was determined by the MTT assay and the variation in PGC1A expression suggests a direct impact on mitochondrial function/biogenesis, which should be further investigated.

It has been reported that *HSPB1* (HSP27) can act as a tumor suppressor [58], and its downregulation has been reported as a therapeutic target for prostate cancer [59]. Here, both the T4 and M13 phages have a negative effect on *HSPB1* (HSP27) in the first 24 h of interaction with the LNCaP cell line. Therefore, *AR*, *PGC1A* and *HSPB1* (HSP27) down-regulation by phages exposures can be explored as potential strategies for prostate cancer therapies [52,53,60,61].

## 5. Conclusions

Bacteriophages M13 and T4 interact with mammalian cells and induce remarkable gene expression changes in LNCaP cancer cells. It is becoming apparent that phage interaction with cancer cells affects both cell metabolism and direct gene expression. The observed pattern of gene expression suggests that cells are less dependent on the AR signaling pathway and more dependent on the *PIK*/*AKT*/*MAPK* pathways, making the cells more vulnerable to existing therapies targeting the *PI3K*/*AKT*/*MAPK* pathways inhibitors. Further studies are necessary to understand how M13 and T4 phages interact with LNCaP cells, as well as to investigate if they make the cells more susceptible to the immune system.

## Figures and Tables

**Figure 1 viruses-13-01754-f001:**
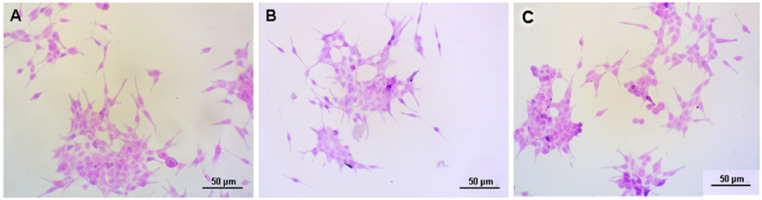
Representative images of LNCaP cells stained with Hematoxylin-Eosin. (**A**) untreated LNCaP cells. (**B**) LNCaP cells exposed to bacteriophage M13 at 1 × 10^7^ pfu/mL. (**C**) LNCaP cells exposed to bacteriophage T4 at 1 × 10^7^ pfu/mL. No significant morphological difference was observed between the treated and untreated cells after 24 h of treatment. Scale bars = 50 µm.

**Figure 2 viruses-13-01754-f002:**
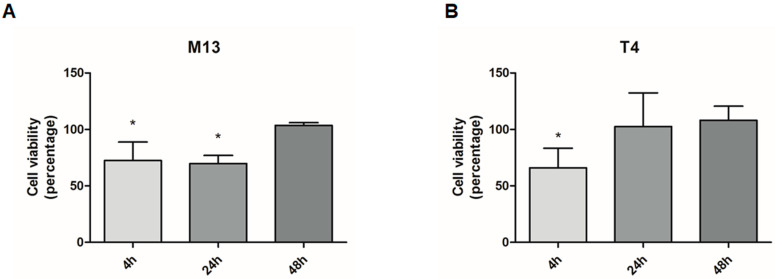
Viability of prostate cell line LNCaP after exposure to 10^7^ pfu/mL of bacteriophages M13 and T4 for 4, 24, and 48 h. (**A**) The viability of LNCaP cells after treatment with M13 decreased significantly at 4 and 24 h (*p* < 0.05). (**B**) LNCaP cell viability after T4 exposition showed a significant reduction at 4 h (*p* < 0.05). Data are expressed in percentage related to untreated cells. * *p* < 0.05 vs. the control group within the same period of observation.

**Figure 3 viruses-13-01754-f003:**
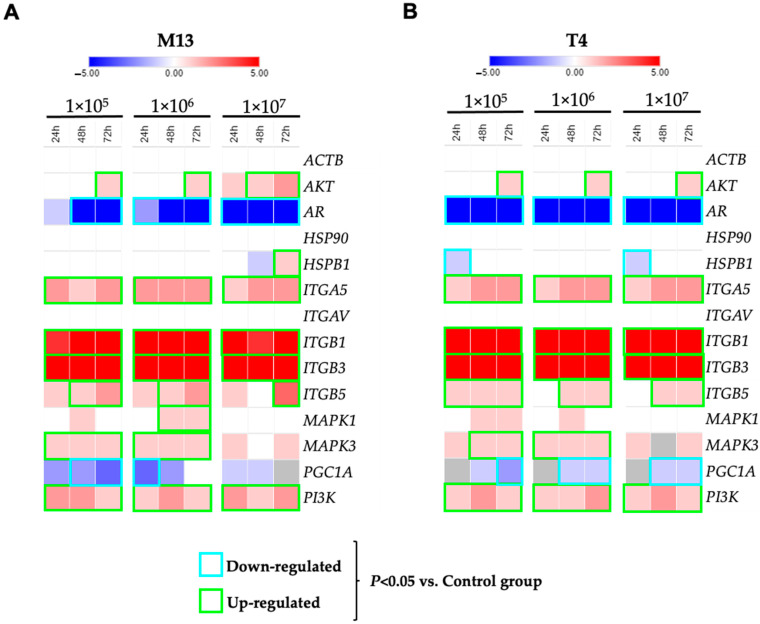
Gene expression of prostate cell line LNCaP after 24, 48, and 72 h of treatment with bacteriophages M13 and T4. Heatmap with a representation of effect on LNCaP cells gene expression after interaction with bacteriophage M13 (**A**) and bacteriophage T4 (**B**). Relative levels of gene expression (median) are shown as a Log_2_Fold-Change. Values between −5 and 0 represent negatively regulated genes (blue gradient) and, between 0 and 5, represent genes with positive regulation (red gradient). The data surrounded green represents the up-regulated gene with *p* < 0.05. The data in blue represents the down-regulated gene with *p* < 0.05.

**Table 1 viruses-13-01754-t001:** Primers used in the RT-qPCR reactions.

Genes	Primer Sense	Primer Anti-Sense
*ACTB*	GATTCCTATGTGGGCGACGA	TGTAGAAGGTGTGGTGCCAG
*AKT*	CATCGCTTCTTTGCCGGTATC	ACTCCATGCTGTCATCTTGGTC
*AR*	GACATGCGTTTGGAGACTGC	CAATCATTTCTGCTGGCGCA
*GAPDH*	GAATGGGCAGCCGTTAGGAA	ATCACCCGGAGGAGAAATCG
*HSP90*	AGGGGGAAAGGGGAGTATCT	ATGTCAACCCTTGGAGCAGC
*HSPB1*	CGCGGAAATACACGCTGCC	GACTCGAAGGTGACTGGGATG
*ITGA5*	GGGTGGTGCTGTCTACCTC	GTGGAGCGCATGCCAAGATG
*ITGAV*	AGGCACCCTCCTTCTGATCC	CTTGGCATAATCTCTATTGCCTGT
*ITGB1*	GCCAAATGGGACACGCAAGA	GTGTTGTGGGATTTGCACGG
*ITGB3*	CTGCCGTGACGAGATTGAGT	CCTTGGGACACTCTGGCTCT
*ITGB5*	GGGCTCTACTCAGTGGTTTCG	GGCTTCCGAAGTCCTCTTTG
*MAPK1*	TCAGCTAACGTTCTGCACCG	ACTTGGTGTAGCCCTTGGA
*MAPK3*	ATCTTCCAGGAGACAGCACG	TTCTAACAGTCTGGCGGGAG
*PGC1A*	GAAGGGTACTTTTCTGCCCCT	CTTCTTCCAGCCTTGGGGAG
*PI3K*	AGAGCCCCGAGCGTTT	TCGTGGAGGCATTGTTCTGA

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
