# Peer review of "Bacteriophages M13 and T4 Increase the Expression of Anchorage-Dependent Survival Pathway Genes and Down Regulate Androgen Receptor Expression in LNCaP Prostate Cell Line"

_viruses, 2021, doi:10.3390/v13091754_

Round 1
Reviewer 1 Report
Sanmukh and coworkers analyzed the change in expression for several genes related to cell adhesion, growth, and survival in an androgen-responsive prostate cancer cell line following treatment with M13 or T4 bacteriophages for 24-72 hours. The results demonstrate significantly modulated gene expression following treatment with either bacteriophage. Given the high demand for bacteriophage-based technology development into medical products, a thorough investigation of the changes in gene expression following their treatment is beneficial to the scientific community.
Major Concerns:
- A detailed description of bacteriophage preparation and purification should be provided. It is not clear if bacteriophage preparations were cleared of bacterial endotoxins, which may present as an artifact to changes in gene expression. Were any downstream TLR or LPS pathways analyzed for changes in gene expression as expected following treatment with phages containing either ssDNA or dsDNA.
- Are the same patterns observed in an androgen-unresponsive prostate cell line? This would strengthen claims for involvement of the PI3K/AKT pathway for therapeutics.
Minor Concerns:
- Line 157 – “The exposure to bacteriophage temporarily reduced the viability of prostate cancer cells” – It does not seem clear that treated cells would experience a decrease in viability and may only be a reduction in proliferation rate. Further experiments would be required to address if changes were due to modification of the proliferation rate, inhibition of cell cycle progression, or decrease in viable cell numbers by programmed cell death pathways.
- Line 178 – Figure 2 – Y-axis should be updated to show the percentage of viable cells as described in the material and methods section. Figure 2 shows cell viability and not the absorbance. Error bars should also be provided. Statistically significant changes should be identified on the plot.
- Line 200 – changes in cell morphology were not apparent. However, no additional verification to changes in protein were demonstrated, i.e. immunofluorescence or Western blotting to clearly show modulation of protein expression and changes in cell morphology.
Author Response
Reviewer 1:
Sanmukh and coworkers analyzed the change in expression for several genes related to cell adhesion, growth, and survival in an androgen-responsive prostate cancer cell line following treatment with M13 or T4 bacteriophages for 24-72 hours. The results demonstrate significantly modulated gene expression following treatment with either bacteriophage. Given the high demand for bacteriophage-based technology development into medical products, a thorough investigation of the changes in gene expression following their treatment is beneficial to the scientific community.
Major Concerns:
- A detailed description of bacteriophage preparation and purification should be provided. It is not clear if bacteriophage preparations were cleared of bacterial endotoxins, which may present as an artifact to changes in gene expression. Were any downstream TLR or LPS pathways analyzed for changes in gene expression as expected following treatment with phages containing either ssDNA or dsDNA.
Response: This is a very interesting question. We bought the phages from the manufacturers with a certain degree of purification and at higher concentrations of phages 1011. In our experiments, we did not need to expand the phages in bacterial cultures. We also performed one centrifugation and filtering step. But, most importantly, we performed several steps of dilutions to reach 105, 106, 107 for use as final treatment concentration. We believe that these serial dilutions inimized any trace of contaminations. These details have been included in the M&M section.
We did not analyse any downstream TLR or LPS pathways in our manuscript. We do agree with the Reviewer that the observed changes in gene expression profile might arise from these effects. However, this was not the focus of our manuscript. As other articles already have some information about this, we have included comments on the introduction about the presence/influence of phage DNA on cells and expression patterns.
About TLR and LPS effects on Cancer cells, we addressed this issue and observed no effect on cell viability and growth for this cell line. Besides this cell present several TLR subtypes, the activation of TLR in this cell appears to not be as strong as in immune system cells.
Rezania et al. The same and not the same: heterogeneous functional activation of prostate tumor cells by TLR ligation. Cancer Cell Int 14, 54 (2014). https://doi.org/10.1186/1475-2867-14-54
- Are the same patterns observed in an androgen-unresponsive prostate cell line? This would strengthen claims for involvement of the PI3K/AKT pathway for therapeutics.
Response: This is an interesting question. We have tested the effects of phages on the androgen independent PC3 cells. The number of genes evaluated was small and we have concentrated on the expression of HSP90 (Sanmukh et al 2017) and Integrins (Submitted manuscript). We have well reported a considerable effect on growth and migration of both LNCaP (androgen dependent) and PC3 (androgen independent) cell lines (Sanmukh and Felisbino, 2018). These effects on LNCaP seem not specific for this cell line.
Sanmukh, S.G.; Felisbino, S.L. Development of pipette tip gap closure migration assay (s-ARU method) for studying semi-adherent cell lines. Cytotechnology 2018, 70, 1685–1695, doi:10.1007/s10616-018-0245-1.
Sanmukh SG, Dos Santos SAA, Felisbino SL. Natural bacteriophages T4 and M13 down-regulates Hsp90 gene expression in human prostate cancer cells (PC-3) representing a potential nanoparticle against cancer. Virol Res J. 2017;1(1):21-23.
Minor Concerns:
- Line 157 – “The exposure to bacteriophage temporarily reduced the viability of prostate cancer cells” – It does not seem clear that treated cells would experience a decrease in viability and may only be a reduction in proliferation rate. Further experiments would be required to address if changes were due to modification of the proliferation rate, inhibition of cell cycle progression, or decrease in viable cell numbers by programmed cell death pathways.
Response: Many thanks for this comment. We have considered this possibility in deep and, on second thought, we believe that the transient decrease in MTT reduction might indeed correspond to a reduction on the mitochondrial capacity to reduce the dye. This is so because we believe there would be no time enough for the restoration of cell viability as observed and because we found changes in the gene of a key regulator of mitochondrial physiology A sentence has been included in the discussion section.
- Line 178 – Figure 2 – Y-axis should be updated to show the percentage of viable cells as described in the material and methods section. Figure 2 shows cell viability and not the absorbance. Error bars should also be provided. Statistically significant changes should be identified on the plot.
Response: The figure has been changed according to the reviewer instructions.
- Line 200 – changes in cell morphology were not apparent. However, no additional verification to changes in protein were demonstrated, i.e. immunofluorescence or Western blotting to clearly show modulation of protein expression and changes in cell morphology.
Response: We could not make additional experiments, particularly due to restrictions imposed by the COVID-19 pandemics and quarantine restrictions. However, our data showed no variation in the expression of ACTB, an important cytoskeletal gene, suggesting the observed changes did not include significant changes in the cytoskeletal components. We do agree that additional experiments should be done to highlight the possible effect on cytoskeletal organization.

Reviewer 2 Report
The Authors describe the role of M13 and T4 bacteriophages in the modulation of the expression of genes related to cell adhesion, growth, and survival of LNCaP prostatic adenocarcinoma-derived epithelial cell line. Although the manuscript seems to be interesting, the rational such as the obtained results and their discussion are not sufficient to recommend the publication. I feel that the Authors need to provide more details to allow appropriate interpretation by the reader, and should be more careful with their conclusions not fully supported by the results.
Major revision:
Introduction
- It looks very thin.
- Authors should extend the discussion about bacteriophage, which appears too synthetic with references added without an apparent logical line. I suggest to expand the concept about engineered phages used in this prospective using the reference already present.
- Authors should also include references about multivalent architecture of landscape phage displayed proteins [some examples https://doi.org/10.3390/v11110988, https://doi.org/10.1093/nar/gkaa1279] which affect the biodistribution of phage clones able to specifically recognize non-bacterial targets, such as neoplastic ones.
- What is the rationale of the work?Why do the Authors think that viruses that naturally infect bacteria (bacteriophages) should affect cancer cells?
- In line 62-63, Authors write "it is essential to know how non-modified phages present normally in our body affect cancer cells". What do they mean by "present normally in our body".
- Similar to above, what is the rational to use both lytic and lysogenic bacteriophage?
Materials and Methods
- When it first appears, the name line should be followed by the atcc number (which should be ATCC CRL-1740).
- Similar to above, also M13 and T4 names should be added with collection code from New England Biolabs Inc and Carolina Inc, respectively.
- Line 90, How was the phage/cell ratio chosen? Did the ss carry out preliminary tests?
- Line 92, Phosphate buffered saline should be written in full the first time, including the abbreviation PBS below.
- About Cell Viability Assay, Authors should shortlydescribe the protocol used since the two protocols differ from each other, in terms of incubation time, buffer to dissolve formazan, reading wavelength (Authors use 550nm, while in the reference has been carried out at 570nm).
- The above consideration must be valid for all other protocols not described. It is always advisable to give a brief description of the reference protocols and / or any modifications to allow the reader to use and adapt these protocols to his experimental needs.
Results
- At what time was the morphological evaluation done? Figure 1 should include all time data, i.e., 24, 48 and 72 hours.
- The graph of Figure 2 should be replaced with a histogram graph, showing on the y axis the cell viability as percentage from the ratio, at each incubation time, between the absorbance of the condition with the phage and that of the negative control multiplied byone hundred.
- In addition, in Figure 2 the data of the 72 hours are missing.
Discussion
- Line 203-204, How do the Authors deduce that phage is internalized in cells?The reduction of vitality could be given by an external action of the phages on the cancer cell. Otherwise, the Authors should show any data that verify the internalization (such as described in https://doi.org/10.3390/v11090785).
- How the Authors explain the different variation in the viability from 24h to 48h under M13 treatment? Are there the connections with gene expression?
- Although not significant, the increase in viability of LNCaP with T4 at 48h appears interesting, even from the comparison with the results at 4 h.The Authors have some explanation for this, supported by the gene expression results.
- Do the Authors think that the bacteriophages are specific to LNCaP cells?
Other:
- 50% of the references are prior to 2015
Author Response
Reviewer 2.
The Authors describe the role of M13 and T4 bacteriophages in the modulation of the expression of genes related to cell adhesion, growth, and survival of LNCaP prostatic adenocarcinoma-derived epithelial cell line. Although the manuscript seems to be interesting, the rationale such as the obtained results and their discussion are not sufficient to recommend the publication. I feel that the Authors need to provide more details to allow appropriate interpretation by the reader and should be more careful with their conclusions not fully supported by the results.
Introduction
- It looks very thin.
Response: As per reviewers’ suggestion significant changes are made in the introduction. New references are referred to and cited to support the rationale behind our experimental design as well as works supporting our results.
- Authors should extend the discussion about bacteriophage, which appears too synthetic with references added without an apparent logical line. I suggest expanding the concept about engineered phages used in this prospective using the reference already present.
Response: As above mentioned, the introduction has been extended and this specific topic included.
- Authors should also include references about multivalent architecture of landscape phage displayed proteins [some examples https://doi.org/10.3390/v11110988, https://doi.org/10.1093/nar/gkaa1279] which affect the biodistribution of phage clones able to specifically recognize non-bacterial targets, such as neoplastic ones.
Response: The references had been included in the article, as per the reviewer's suggestion.
- What is the rationale of the work? Why do the Authors think that viruses that naturally infect bacteria (bacteriophages) should affect cancer cells?
Response: This is an interesting question. In the context of prostate diseases, bacteriophages has been proposed as a therapeutic approach for treating bacterial prostatitis. In this sense the interaction of phages with normal and tumoral prostatic cells appears inevitable. Moreover, since we have previously reported how phages affect the growth and migration of LNCaP (androgen dependent) and PC3 (androgen independent) cell lines (Sanmukh et al. 2017; Sanmukh and Felisbino,2018), it appears mandatory, in the present manuscript, to understand the effect of phages on the morphology and gene expression in tumoral cancer cells.
Sanmukh, S.G.; Felisbino, S.L. Development of pipette tip gap closure migration assay (s-ARU method) for studying semi-adherent cell lines. Cytotechnology 2018, 70, 1685–1695, doi:10.1007/s10616-018-0245-1.
Sanmukh SG, Dos Santos SAA, Felisbino SL. Natural bacteriophages T4 and M13 down-regulates Hsp90 gene expression in human prostate cancer cells (PC-3) representing a potential nanoparticle against cancer. Virol Res J. 2017;1(1):21-23.
- In line 62-63, Authors write "it is essential to know how non-modified phages present normally in our body affect cancer cells". What do they mean by "present normally in our body".
Response: This paragraph has been modified for the sake of clarity. We meant phages infect bacteria present in our intestinal microbiota.
Bodner, K., Melkonian, A. L., & Covert, M. W. (2021). The Enemy of My Enemy: New Insights Regarding Bacteriophage-Mammalian Cell Interactions. Trends in microbiology, 29(6), 528–541. https://doi.org/10.1016/j.tim.2020.10.014
- Similar to above, what is the rationale to use both lytic and lysogenic bacteriophage?
Response: As above mentioned, both phages were found to promote changes in gene expression in other cell types. Here we extend these observations, testing their effects on LNCaP cells. The reason for using M13 and T4 phages was due to their particle size which is mostly proteinaceous in nature, and our objective was to study the effect of different types of phages (filamentous and non filamentous phage particles) on the cancer cell growth and migration initially. The lytic and lysogenic nature of our phages under consideration was never a rationale for this work, but different phage dimensions with their coat proteins were our main interests. As the number of peptide molecules present on the phages varies for different phages, the effect of natural phage peptides can be easily demonstrated as can be observed from our reported MTT assays as well as gene expression studies.
Materials and Methods
- When it first appears, the name line should be followed by the atcc number (which should be ATCC CRL-1740).
Response: The ATCC number has been included in the Materials and Method section as per reviewers suggestion.
- Similar to above, also M13 and T4 names should be added with collection code from New England Biolabs Inc and Carolina Inc, respectively.
Response: The collection code names for M13 phage: M13KE Phage and T4 phage: Coliphage T4 (T4r+) have been added as per reviewers suggestions.
- Line 90, How was the phage/cell ratio chosen? Did the ss carry out preliminary tests?
Response: This is a very interesting point of our study. We did preliminary tests and now, in this version of the manuscript, we present results of three different concentrations of phages in cell culture medium. The corresponding results have been added to the manuscript. A very important research group in the field of bacteriophage, Dabrowska et al., 2009, used a high concentration of phage 109 pfu to study their effect on migration. But in our case, we were interested in checking the effect of a lower phage concentration on different cancer progression genes as well as viability.
Da̧browska et al. The effect of bacteriophages T4 and HAP1 on in vitro melanoma migration. BMC Microbiol. 2009, 9, doi:10.1186/1471-2180-9-13.
- Line 92, Phosphate buffered saline should be written in full the first time, including the abbreviation PBS below.
Response: The changes are made for the Phosphate Buffered Saline abbreviation, as per the reviewer's suggestion
- About Cell Viability Assay, Authors should shortly describe the protocol used since the two protocols differ from each other, in terms of incubation time, buffer to dissolve formazan, reading wavelength (Authors use 550 nm, while in the reference has been carried out at 570 nm).
Response: The authors thank the reviewer for the observation. The MTT protocol was described in detail, and now appears as:
- The above consideration must be valid for all other protocols not described. It is always advisable to give a brief description of the reference protocols and / or any modifications to allow the reader to use and adapt these protocols to his experimental needs.
Response: Protocols were revised for the sake of clarity, as suggested.
Results
- At what time was the morphological evaluation done? Figure 1 should include all time data, i.e., 24, 48 and 72 hours.
Response: The morphological evaluation was done after 24, 48 and 72 hours of phage exposure at 1x107 concentration. No significant changes were also observed after 24, 48 and 72 hours. So we just present 24 hours as representative of no morphological changes. A sentence has been included in the Material and Methods and Results sections.
- The graph of Figure 2 should be replaced with a histogram graph, showing on the y axis the cell viability as percentage from the ratio, at each incubation time, between the absorbance of the condition with the phage and that of the negative control multiplied by one hundred.
Response: Graphs were modified accordingly.
- In addition, in Figure 2 the data of the 72 hours are missing.
Response: The figure and its legend were corrected as per the reviewer's suggestion. We have also modified the figure to have a better understanding about the individual effect of two phages on LNCaP cell viability. Since at 48 hours the cell viability of both treatments were similar to untreated cells, we did not perform an additional 72 hours experiment.
Discussion
- Line 203-204, How do the Authors deduce that phage is internalized in cells? The reduction of vitality could be given by an external action of the phages on the cancer cell. Otherwise, the Authors should show any data that verifies the internalization (such as described in https://doi.org/10.3390/v11090785).
Response: We have removed the statement regarding internalisation of phages as per reviewers comments. Since we are not able to show any data about this process, we have decided to explain in detail about the development of phage internalisation in the introduction section using suitable literature.
- How the Authors explain the different variation in the viability from 24h to 48h under M13 treatment? Are there the connections with gene expression?
Response: This is an interesting question. As explained earlier, the phage M13 dimensions seems to affect the viability of LNCaP cancer cell line stronger than T4. As observed by the reviewer, we think that early down regulation of androgen receptor gene could be involved with the initial reduction of cell viability, which induces a shift/adaptation in cellular mechanism of survival and growth to the PI3K/AKT/MAPK pathway.
- Although not significant, the increase in viability of LNCaP with T4 at 48h appears interesting, even from the comparison with the results at 4 h. The Authors have some explanation for this, supported by the gene expression results.
Response: This is an intriguing point. As explained earlier, the phage dimensions seem to affect the viability of LNCaP cancer cell line differently due to their varying sizes. As pointed out in the previous answer, we think that early down regulation of androgen receptor gene could be involved with the initial reduction of cell viability, which induces a shift/adaptation in cellular mechanism of survival and growth to the PI3K/AKT/MAPK pathway.
- Do the Authors think that the bacteriophages are specific to LNCaP cells?
Response: We don't think that the bacteriophages are specific to LNCaP cells as we have already observed such non-specific binding in PC3 prostate cancer cell lines previously reported by us (Sanmukh and Felisbino, 2018) and we have another article under review with PC3 cell lines. We have also shown that bacteriophage M13 and T4 bind to PC3 cells (Sanmukh et al., 2017).
Sanmukh, S.G.; Felisbino, S.L. Development of pipette tip gap closure migration assay (s-ARU method) for studying semi-adherent cell lines. Cytotechnology 2018, 70, 1685–1695, doi:10.1007/s10616-018-0245-1.
Sanmukh SG, Dos Santos SAA, Felisbino SL. Natural bacteriophages T4 and M13 down-regulates Hsp90 gene expression in human prostate cancer cells (PC-3) representing a potential nanoparticle against cancer. Virol Res J. 2017;1(1):21-23.
Other:
- 50% of the references are prior to 2015
Response: We have included new references suggested by the reviewers as well as others supporting our findings

Reviewer 3 Report
The authors descibe the effect of M13 and T4 bacteriophages on the expression of certain genes in prostate cancer cell line, which could be significant for the ability of invasion of such cells and different signalling pathways indicative of progress on the cancer disease. Regarding the increasing importance of phage-based therapies, this contribution is significant to the scientific community. The experiments are well explained and the reasoning behind the conclusions clear. My comments would mostly refer to the structure of the paper and and the need to address certain points in experimental design.
- The genes observed for their modified regulations should be listed in the Introduction section, grouped according to their function (or the function of the product) and their choice explained. Parts of the Discussion Section could be used there. To that, also design (choice or source) of the primers should be mentioned (in the Material and Methods Section)
- How were pfu units of the phage preparation determined?
- Vehicle (buffer) treatment is stated as the negative control: what volume proportion was the phage preparation and are you sure that this is negligible?
- The same amount of pfu was used for treatment of different number of the cells in different experiments, could you please explain the reason?
- Time dependency of phage treatment is presented, but data on concentration dependency, at least for the strongly differentially affected genes, would improve the statements of the manuscript.
- Authors state that the values are normalized to the effect on GAPDH and ACTB. was there any effect on the reference genes? Were the results for 2 reference genes similar, and if not, how exactly was the normalization with 2 reference genes performed? This is particulary interesting for ACTB if the cytoskeleton is really affected by phage particles.
List of minor remarks, which I hope you will find helpful:
Line 35: I think it is not necessary to describe the negative control in the abstract
Line 43: PCa abbreviation should be prostate cancer here
Line 51: authors later mention the possibility of use of phage in combination therapies, this option could also be listed here
Line 84: …in fresh culture medium oft he same composition as above (I assume)
Line 95: proliferation might be better expression than viability to describe the outcome of the test
Line 138: listed are 15 genes, 13 target and 2 reference. Also Figure presents the results for 13 target genes.
Line 243: valuable might be a better expression that „influential“
Line 247: transient decrease in cell proliferation as determined with the MTT assay
Line 254: potential strategy instead of potential „target“
Author Response
Reviewer 3
The authors describe the effect of M13 and T4 bacteriophages on the expression of certain genes in prostate cancer cell line, which could be significant for the ability of invasion of such cells and different signalling pathways indicative of progress on the cancer disease. Regarding the increasing importance of phage-based therapies, this contribution is significant to the scientific community. The experiments are well explained and the reasoning behind the conclusions clear.
My comments would mostly refer to the structure of the paper and the need to address certain points in experimental design.
- The genes observed for their modified regulations should be listed in the Introduction section, grouped according to their function (or the function of the product) and their choice explained. Parts of the Discussion Section could be used there. To that, also design (choice or source) of the primers should be mentioned (in the Material and Methods Section)
Response. The authors thank the observation of the reviewer. The genes which were used in these studies are explained in the introduction section according to their function and their effect as per reviewers’ suggestion and in the materials and method section. Similarly, the reasons for their selection are also explained with the results obtained.
- How were pfu units of the phage preparation determined?
Response: This information was missing in the manuscript. The PFU of both the phages were determined for the original stock obtained from the manufacturer as per their specific guidelines. Also, as the following, the determination of titer, the phage dilutions were prepared in PBS for the experiments. Phage enrichment was avoided to prevent bacterial contamination as well as endotoxin production due to bacterial debris and phage lytic activities. A sentence has been included in the material and Methods section
- Vehicle (buffer) treatment is stated as the negative control: what volume proportion was the phage preparation and are you sure that this is negligible?
Response: All the dilutions were prepared by using PBS and as the original stocks of procured phages were diluted over 100 folds to get the concentration of 109 PFU/ml. From this 10 µl was added in 1 ml of culture medium for the treatments at 107 PFU/ml. So we believe that 10 ul of PBS do not compromise the culture medium formula in both treated and untreated cells.
The same amount of pfu was used for treatment of different numbers of the cells in different experiments, could you please explain the reason?
Response: This is an important information that was not included previously in the manuscript. We have now included the results obtained for 3 different phage concentrations (105, 106, 107 PFU) used for the LNCaP treatment and 3 different time periods of treatments ( 24, 48 and 72 hours) for our reported experiments. And we have obtained very similar results. So we think the cell number is not interfering in our results. Moreover, we have tried to work with cells around 70-80% of confluency. So we believe even a small amount of phages is capable of inducing strong responses in mammalian cells.
- Time dependency of phage treatment is presented, but data on concentration dependency, at least for the strongly differentially affected genes, would improve the statements of the manuscript.
Response: The authors strongly agree with the reviewer that this data was missing. We have included the phage concentration dependent gene expression changes associated within the LNCaP cell line. Also, changes have been made in the materials and methods section and explanations regarding the findings have been included in the results and discussion section.
- Authors state that the values are normalized to the effect on GAPDH and ACTB. Was there any effect on the reference genes? Were the results for 2 reference genes similar, and if not, how exactly was the normalization with 2 reference genes performed? This is particularly interesting for ACTB if the cytoskeleton is really affected by phage particles.
Response: Generally, our lab uses at least three endogenous controls for RT-PCR: TBP, GAPDH and ACTB. And we usually perform geometric mean/average of delta Ct of two or three house-keeping genes. An explanation of this can be obtained in this work (https://www.biorxiv.org/content/10.1101/2020.01.29.925651v1.full.pdf). Initially, we performed ACTB as a second house-keeping gene in case of the GAPDH variations. However, as well observed by the reviewers that ACTB could be a target gene for cytoskeleton effect of phage interaction, we decided to use just GAPDH as a house-keeping gene. The integrin pathway affected by phage interaction might mobilise actin protein from cytoskeleton, but without changes in its gene expression. This aspect will certainly be an issue for further studies in our lab.
List of minor remarks, which I hope you will find helpful:
Line 35: I think it is not necessary to describe the negative control in the abstract
Response: The negative control is removed as per reviewers comments
Line 43: PCa abbreviation should be prostate cancer here
Response: The changes were made in the article as per reviewers suggestion
Line 51: authors later mention the possibility of use of phage in combination therapies, this option could also be listed here
Response: The changes are made as per reviewers suggestions
Line 84: …in fresh culture medium of the same composition as above (I assume)
Response: The statement is included in the article as per suggestion
Line 95: proliferation might be better expression than viability to describe the outcome of the test
Response: The statement is included in the article as per reviewers suggestion
Line 138: listed are 15 genes, 13 target and 2 reference. Also Figure presents the results for 13 target genes.
Response: Corrected accordingly. But now we included ACTB as a target gene and left GAPDH as reference.
Line 243: valuable might be a better expression that "influential“
Response: The statement was corrected as per reviewers suggestion.
Line 247: transient decrease in cell proliferation as determined with the MTT assay
Response: The statement was corrected as per reviewers suggestion.
Line 254: potential strategy instead of potential „target“
Response: The statement was corrected as per reviewers suggestion

Round 2
Reviewer 1 Report
Sanmukh and coworkers have addressed the concerns and have significantly improved the overall quality of the manuscript.
Minor changes in language are recommended to improve overall readability. For example, but not limited to:
- Line 51 – “Bacteriophages have been reported to utilize mammalian viruses route of infections for their entry into the cells…”
- Line 55 – “Moreover, phages DNA is reported to activate the expression…”
- Line 106 – “The phages were diluted 100x to 109 in phosphate buffer saline…” -> Phage preparations were diluted 100-fold with phosphate buffered saline (1X PBS) to reach a concentration of ~109 pfu/mL.
Author Response
Sanmukh and coworkers have addressed the concerns and have significantly improved the overall quality of the manuscript. Minor changes in language are recommended to improve overall readability. For example, but not limited to: •
Line 51 “Bacteriophages have been reported to utilize mammalian viruses route of infections for their entry into the cells…”
Response: The changes are made accordingly as per reviewer suggestion. •
Line 55 “Moreover, phages DNA is reported to activate the express ion…” Response: The corrections are made to the sentence as per reviewer suggestion. • Line 106 “The phages were diluted 100x to 10 9 in phosphate buffer saline…” preparations were diluted 100-- > Phage fold with phosphate buffered saline (1X PBS) to reach concentration of ~10 9 pfu/mL. a •
Response: The changes are made accordingly as per reviewer suggestion.
